# **Assessing the Detection of Methane Plumes in Offshore Areas Using High-Resolution Imaging Spectrometers**

Javier Roger<sup>1</sup>, Luis Guanter<sup>1,2</sup>, and Javier Gorroño<sup>1</sup>

<sup>1</sup>Research Institute of Water and Environmental Engineering (IIAMA), Universitat Politècnica de València (UPV), 46022, Valencia, Spain.

<sup>2</sup>Environmental Defense Fund, Reguliersgracht 79, 1017 LN Amsterdam, The Netherlands.

**Correspondence:** Javier Roger (jarojua@upv.es)

## Abstract.

The offshore oil and gas industry is an important contributor to global anthropogenic methane emissions. Satellite-based, high-resolution imaging spectrometers are showing a great potential for the detection of methane emissions over land. However, the use of the same methods over offshore oil and gas extraction basins is challenged by the low reflectance of water in the near- and shortwave infrared spectral windows used for methane retrievals. This limitation can be partly alleviated by data acquisitions under the so-called sun glint configuration, which enhances the at-sensor radiance. In this work, we assess the performance of two space-based imaging spectrometers, EnMAP and EMIT, for the detection of offshore methane plumes applying the matched filter method. We use simulated plumes to generate parametric probability of detection (POD) models for a range of emission flux rates (Q), at-sensor radiances and wind speeds. The POD models were confronted with real plume detections for the two instruments. Our analysis shows that the spatial resolution of the instrument and the at-sensor radiance (which drives the retrieval precision) are the two factors with the greatest impact on plume POD. We also evaluate the dependence of the at-sensor radiance on the illumination-observation geometry and the surface roughness. Our POD models properly represent the different trade-offs between spatial resolution and retrieval precision in EnMAP and EMIT. As an example, for most combinations of Q and wind speed values at POD = 50 %, EMIT demonstrates better detection performance at Q > 7 t/h, whereas EnMAP performs better at Q < 7 t/h. This study demonstrates the ability of these two satellite instruments to detect high-emitting offshore point sources under a range of different conditions. By filtering data based on these conditions, methane emission detection and monitoring efforts can be optimized, reducing unnecessary searches and ultimately increasing the action taken on these emissions.

#### 1 Introduction

Mitigation of methane emissions from anthropogenic sources is key to curb climate change (UNEP, 2021). The oil and gas (O&G) industry, which accounts for around 25 % of anthropogenic emissions, is an important sector in this context, as a large proportion of emissions can be mitigated cost-effectively (Ocko et al., 2021). Within this sector, offshore platforms account for approximately 28 % of the total production (statista, 2024a, b). Atmospheric measurement is crucial to detect and monitor emissions in different areas, and to pinpoint those sites where mitigation is most needed.

Remote sensing using space-based sensors has proven instrumental in detecting methane emissions over land, although the typically low reflectance of water presents a challenge for emission detection and quantification in offshore areas (Roger et al., 2024). A low reflectance results in acquisitions with low levels of radiance. This leads to noisy methane retrievals, where emissions cannot be distinguished. However, an increase in observed radiance can be obtained by leveraging the sun glint effect, which occurs when the angular configuration between the instrument and the sun is set close to a mirror-like configuration. Acquisitions that meet (or closely meet) this condition will result in more favorable retrievals for methane emission detection (Ayasse et al., 2018).

25

40

Among the satellite-based instruments that have been successfully used to detect and quantify methane emissions, are the multispectral missions such as Landsat-8/9 (L8/9) (Irakulis-Loitxate et al., 2022b, a; Jacob et al., 2022), and Sentinel-2 (S2) (Varon et al., 2021; Gorroño et al., 2023). They typically have a coarse spectral resolution and sampling, and a ground sampling distance (GSD) of 20-30 m. These instruments follow fixed orbits and therefore cannot point to improve the angular configuration between the sun and the sensor to obtain close-to-sun glint acquisitions. However, their large swath and the high temporal resolution results in a better ability for target monitoring. The WorldView3 (WV3) mission (Sánchez-García et al., 2022) is another multispectral instrument, but it has a better ground sampling distance of 4 m and it is capable of pointing. Nevertheless, it is a commercial mission and their products are not freely available.

In addition, hyperspectral missions such as PRISMA (Guanter et al., 2021), EnMAP (Roger et al., 2024), and EMIT (Thorpe et al., 2023) have shown a high sensitivity to methane. These instruments have a high spectral resolution and sampling around 10 nm, but exhibit a low temporal resolution and a ground sampling distance of 30 m, except for EMIT with a ground sampling distance of 60 m. Despite the lower spatial resolution, EMIT demonstrates a remarkable signal-to-noise-ratio (SNR). PRISMA and EnMAP instruments can point in the across-track direction in a  $\pm 21^{\circ}$  and  $\pm 30^{\circ}$  range, respectively, and target acquisitions can be tasked by the user. Additionally, there is the commercial GHGSat constellation (Varon et al., 2019; Jervis et al., 2021), based on wide-angle Fabry-Perot spectrometers with a ground sampling distance of 20-50 m at nadir and a very fine spectral resolution. It has the ability to point in the across ( $\pm 55^{\circ}$ ) and along-track ( $\pm 65^{\circ}$ ) direction, which translates to a high flexibility to obtain close-to-sun glint acquisitions (MacLean et al., 2024).

Several campaigns for detecting offshore emissions have been carried out using ship (Nara et al., 2014; Riddick et al., 2019; Hensen et al., 2019; Yacovitch and Daube, 2020) and airborne (Gorchov Negron et al., 2020; Foulds et al., 2022; Ayasse et al., 2022; Negron et al., 2023, 2024) instruments, covering areas in various parts of the world such as the North Sea, the US Gulf of Mexico (GoM), and Southeast Asia. As a result, distributions of flux rate (Q in t/h) values have been obtained for different offshore areas. Median Q values from these distributions range approximately between 0.01 t/h and 0.36 t/h, which currently poses a challenge for the minimum detection capabilities of most satellite-based instruments (Jacob et al., 2016; Cusworth et al., 2019; Guanter et al., 2021; Thorpe et al., 2023). However, it has been found that these distributions typically exhibit a pronounced skewness that leads to stronger emissions contributing significantly to the overall distribution budget. For instance, in an airborne campaign conducted in the U.S. Gulf of Mexico, Ayasse et al. (2022) showed that 11 % of the sampled sources, each emitting at a rate of Q > 1 t/h, accounted for 50 % of the total emissions detected. The presence of large emitting sources

increases the usefulness of satellite-based instruments for detection and quantification. Uncovering the highest emissions will play a crucial role in understanding the total amount of emitted methane in offshore sites.

Several studies have highlighted the detection of methane plumes from offshore O&G areas using different satellite instruments. Irakulis-Loitxate et al. (2022a) reported emissions from an ultra-emitter (92-111 t/h) in the Mexican GoM using L8 and WV3. Valverde et al. (2023) showed plumes from a platform in the Gulf of Thailand with S2 (23 t/h) and PRISMA (5 t/h). Moreover, Roger et al. (2024) detected two emissions (both 1 t/h) from a single EnMAP acquisition over the U.S. GoM, while MacLean et al. (2024) presented emissions observed in different parts of the world as low as 0.18 t/h using GHGSat data. There are also online portals such as the NASA's JPL portal (JPL, 2024), that displays several offshore plumes captured by the EMIT instrument. Similarly, the UNEP's IMEO Methane Data portal (IMEO, 2024) showcases methane emissions detected by various satellites, which are gradually contributing to a comprehensive global inventory, including emissions originating from offshore sites.

In this work, we aim to assess the capability of methane emissions detection from offshore areas using the EnMAP and EMIT data. Both provide open data from satellite-based sensors with high sensitivity to methane and have already proven their ability to detect offshore emissions. Moreover, given the strong similarities between the EnMAP and PRISMA instruments, we also aim to approximately infer PRISMA performance using the results extracted from EnMAP data. Results from this study will advance the current understanding of the strengths and limitations of methane emission detection from space in offshore areas, which will contribute to more efficient use of data, optimize detection and monitoring of offshore emissions, and ultimately increase the action taken on methane emissions.

## 2 Materials and methods

85

#### 2.1 Methane emission detection and quantification

Methane column concentration enhancement maps ( $\Delta$ XCH<sub>4</sub>) are generated using the approach described in Thompson et al. (2016), where the matched-filter method is applied in the 2300 nm methane absorption window and an integration over an 8-km high column is assumed. We additionally account for the matched-filter sparsity assumption (Foote et al., 2020). First, we identify those pixels that exceed 2 retrieval standard deviation ( $2\sigma_{\Delta XCH_4}$ ) from an initial iteration. Then, we exclude these pixels for the calculation of the mean and covariance matrix in a second iteration. In this manner, we exclude those pixels related to artifacts and potential methane emissions since  $\sigma_{\Delta XCH_4}$  represents the retrieval background noise (Guanter et al., 2021), i.e. retrieval error.

Plume detection is applied in the  $\Delta$ XCH<sub>4</sub> maps, measured in parts-per-million (ppm), using an emission detection algorithm as described in Gorroño et al. (2023). First, a median filter with a 3x3 kernel is applied to the retrieval to remove the high-frequency noise. Then, we obtain a mask by keeping only those pixels from the filtered retrieval with values greater than a  $2\sigma_{\Delta XCH_4}$  threshold. These are The number of pixels considered to deduce  $\sigma_{\Delta XCH_4}$  is so large that the presence of the plume has little effect. Therefore, background pixels are excluded and only the potential pixels containing methane emission enhancements. To consider that are considered. To evaluate whether a plume is detected, an observation of a plume-like shape

in the  $\Delta XCH_4$  maps is generally required. This condition can only be met with a minimum number of pixels (N) that depends on the instrument resolution and whether a conservative or more relaxed criterion is applied. Then, a filtering based on N and other morphological parameters (van der Walt et al., 2014) is applied to the mask, retaining only those clusters with potential plume-like shapes. Using low N values may result in the appearance of more retrieval artifacts, although smaller plumes can still be detected. On the other hand, higher N values represent a more conservative selection, reducing artifacts but failing to identify smaller plumes. For the EnMAP and EMIT instruments, we empirically found N = 10 as an optimal trade off between a reasonable minimum number of pixels to consider plume detection and the total removal of artifacts. In Gorroño et al. (2023), a value of N = 40 was selected for a conservative plume detection with Sentinel-2 data in onshore areas, while a more relaxed value of N = 20 was used for a supervised detection. In our case, we can decrease this value because we work with data capturing offshore areas, where there is usually a higher degree of spectral homogeneity and a lower occurrence of surface structures that lead to artifacts. In addition, the specific locations where we implement the simulated plumes are carefully selected by visual inspection to minimize the disturbance of these factors. Since EnMAP and EMIT have a ground sampling distance of 30 m and 60 m, respectively, the equivalent area to N = 10 pixels differs for both instruments. As a result, the EMIT minimum area of detection (36000 m<sup>2</sup>) is 4 times higher than the one of EnMAP (9000 m<sup>2</sup>). We will account for this point further in the discussion. We also note that other retrieval methodologies different from the one implemented in this work might lead to different results under the same detection algorithm. However, we use the matched filter since it is a state-of-the-art method that has been extensively used in the literature.

The flux rate value is obtained as in Varon et al. (2018) using the following expressions

$$110 \quad Q = \frac{\text{IME} \cdot U_{\text{eff}} \cdot 3.6}{L} \tag{1}$$

where IME (kg) is the total excess of methane (Frankenberg et al., 2016), L (m) is the square root of the area containing the plume, and  $U_{eff}$  (m/s) is the effective wind speed obtained from a calibration done with the Weather and Research Forecasting Model in large-eddy simulation mode (WRF-LES) plumes adapted to the specific instrument resolution and with an associated wind speed at 10 m above surface ( $U_{10}$ ).

$$U_{\text{eff}} = a \cdot U_{10} + b$$
 (2)

where a and b are the resulting calibration coefficients. For PRISMA and EnMAP, we use a = 0.34 and b = 0.44 (Guanter et al., 2021; Roger et al., 2024), and for EMIT we use a = 0.31 and b = 0.4 (Guanter et al., 2024). The U<sub>10</sub> associated to the acquisition measurement time and emission location is obtained from the GEOS-FP reanalysis product (Molod et al., 2012).

#### 2.2 Potential factors affecting methane emission detection in offshore areas

The spatial resolution of the instrument influences detection, as smaller pixels are less affected by background contamination and can more accurately capture the plume shape in methane retrievals. In this context, EnMAP and PRISMA (GSD = 30 m)

outperform EMIT (GSD = 60 m). However, retrieval precision also plays a role in detection. We use  $1-\sigma_{\Delta XCH_4}$  to measure it, assuming that the retrieval distribution follows a Gaussian distribution (Guanter et al., 2021). There is a high dependency between the methane retrieval precision from acquisitions capturing offshore areas and radiance (MacLean et al., 2024). Then, assessing those factors that have an influence in radiance can give us an understanding of which are the more suitable conditions for detection. Among these factors, we consider the scattering glint angle (SGA), the incidence angle (IA), the SNR, the wind speed, and the surface roughness.

SGA can be defined as the angle that measures the proximity to the sun glint configuration, where the azimuth angle between the sensor and the sun ( $\phi$ ) is 180 °, and there is a zero difference between the solar zenith angle (SZA) and the viewing zenith angle (VZA) (Capderou, 2014). Lower values of SGA will result in closer-to-sun glint acquisitions, which are expected to provide higher levels of radiance. On the other hand, IA represents half the angle between two paths: one from the sun to the surface and the other from the surface to the sensor (Bréon and Henriot, 2006). According to the Fresnel coefficient ( $\rho$ ), there is a positive correlation between IA and the radiance obtained over water (Cox and Munk, 1954). SGA, IA, and  $\rho$  can be expressed as follows,

130

135 
$$SGA = \arccos(\cos(SZA) \cdot \cos(VZA) - \sin(SZA) \cdot \sin(VZA) \cdot \cos(\phi))$$
 (3)

$$IA = 0.5 \cdot \arccos(\cos(SZA) \cdot \cos(VZA) + \sin(SZA) \cdot \sin(VZA) \cdot \cos(\phi))$$
(4)

$$\rho = 0.5 \cdot \left[ \frac{\sin(\text{IA} - \text{IA}')^2}{\sin(\text{IA} + \text{IA}')^2} + \frac{\tan(\text{IA} - \text{IA}')^2}{\tan(\text{IA} + \text{IA}')^2} \right]$$
 (5)

where IA' =  $\arcsin(\sin(\text{IA}) / m)$  and m is the ratio between the refraction indexes from uncontaminated water (n = 1.338) and  $\sin(n = 1)$ .

The wind speed at acquisition time can also have an impact in radiance by increasing surface roughness. Calm water surfaces produces a very localized and strong sun glint as in the example shown in Fig. 1, where a radiance map from an EMIT acquisition capturing a Red Sea area alongside the related SGA map illustrate the correlation of sun glint and low SGA values. On the other hand, rough waters generate a more diffuse glint that covers a more extended area (Capderou, 2014). This roughness can also be measured with a normalized 1-standard deviation of radiance ( $\sigma_{Rad}/Rad$ ), which we measure using those radiance bands the instrument radiance band located at  $\sim$  2131 nm (Rad). Methane does not present absorption at this wavelength and therefore the radiance values cannot be altered by the presence of emission, while still preserving similar radiometric levels to those of the 2300 nm absorption window used for the retrieval. Additionally, wind-induced waves can transform a flat water surface into a surface where the local incidence axis varies. This can lead to unrealistic SGA and IA values as these parameters are based on the assumption of a flat surface.

The SNR is another important parameter to consider when assessing radiance. The typical low radiance of water is often related to acquisitions in which the instrument noise is not negligible in reference to the amount of signal reaching the detector.

**Figure 1.** Radiance band at 2131 nm (left) and the equivalent scattering glint angle image (right) in a log scale from an EMIT acquisition capturing a Red Sea area on 2024/06/21, where a pronounced sun glint can be observed.

Figure 2. Relationship between the radiance at  $\sim 2130$  nm and the noise-equivalent delta radiance (NEdL) from EMIT (red), and EnMAP (green).

Instruments with higher SNR values will result in less noisy retrievals, positively impacting detection. In order to extract the SNR values, we use an on-board calibrated noise model for EnMAP (Carmona, 2024) and a noise model for EMIT extracted from an on-orbit calibration with vicarious targets. (Thompson, 2024; Thompson et al., 2024). In Fig. 2, we can observe the instrument noise curves, i.e. the noise-equivalent delta radiance (NEdL), for the EMIT and EnMAP instruments around the ~ 2130 nm bands. NEdL increases with radiance due to the photon shot noise, which is proportional to the square root of the signal. Note that noise levels at this wavelength are very similar to those at the wavelengths within the strong methane absorption window where the matched filter is applied. In addition, if we downsample EnMAP data to the EMIT spatial resolution, we will find a reduction in instrument noise, which in turn increases the total SNR. This reduction depends on the instrument noise correlation among adjacent pixels. A simple test was performed on an EnMAP acquisition that captured the Baikal Lake area, finding a reduction of up to 40 % after downsampling (Appendix A). However, due to the complexity of properly disentangling instrument noise from surface variability, we consider this value to be just a rough approximation.

## 2.3 Methane emission probability of detection

The emission detection capability for a specific instrument can be inferred by using the detection limit concept. Some studies in the literature allude to this term ambiguously since they actually refer either to the Minimum Detection Limit (MDL) or to the Probability of Detection (POD) (Ayasse et al., 2024). For specific measurement conditions and set of scene conditions, MDL is the minimum Q value at which a plume can be detected (Frankenberg et al., 2016; Irakulis-Loitxate et al., 2021), while POD is the probability of detecting a plume for a given Q value (Conrad et al., 2023; Bruno et al., 2024). Hereinafter, we will use the detection limit term to allude to the POD concept.

Due to the typical low reflectance of water, the radiance level of an acquisition becomes the most critical factor for methane emission detection in offshore areas. Therefore, the detection limit is mainly driven by radiance, which raises the importance of knowing the relationship between these two magnitudes. For this purpose, we combine L1 data acquisitions capturing offshore areas (Table C1 and C2) and atmospheric transport simulations. This allows us to generate realistic scenarios containing methane emissions under different conditions, which are useful to characterize the detection limit.

Our synthetic plumes are generated from WRF-LES simulations, which are then transformed into  $\Delta$ XCH<sub>4</sub> maps adapted to the instrument spatial resolution. Note that the Q value linked to these plumes can be changed by simply scaling the  $\Delta$ XCH<sub>4</sub> maps. Then, to assess the detectability of each plume related to a given Q value, we follow the steps illustrated in the diagram shown in Fig. 3. A plume is integrated into a-each one of the subsets from L1 acquisition data (Table C1 and C2) as in Guanter et al. (2021) to replicate real-like emissions. These subsets are carefully selected by visual inspection to minimize the disturbance of surface structures that can lead to artifacts. Within each subset, we did not notice a significant change in the scattering glint angle parameter that would affect the emission detection. Then, we just implemented the synthetic plume in the upper part of the subset. As shown later in Section 3, the whole subset selection provides a wide range of scattering glint angle values, which is valuable to understand the detection performance in reference to the sun glint configuration. Next, we obtain the related methane retrieval and we next apply the detection algorithm (Section 2.1.) to test whether this plume is detected.

If not, we increase 0.1 t/h to the previously considered Q and repeat the process iteratively until the plume is detected. The Q related to this plume can then be considered as the minimum flux rate at which the plume can be detected  $(Q_{min})$ .

Figure 3. Diagram showing the steps followed to obtain the minimum flux rate value  $(Q_{min})$  at which a plume can be detected according to the detection algorithm described in Section 2.1.

The calculation of the  $Q_{min}$  value considers all the acquisitions from Table C1 and C2, and is applied to every plume from our simulation dataset. We consider  $\frac{11-7}{2}$  groups of simulated plumes (see Table 1), each group presenting 25 plumes related to the same  $U_{10}$  value, but showing different plume shapes. Although there may be differences in the  $U_{10}$  values among plumes within the same group, these variations are very small, allowing us to use the mean value to represent the whole group. On the other hand, each acquisition will be characterized by its associated level of radiance. We measure it using the previously defined Rad parameter, which represents radiance at  $C_{\infty}$  2131 nm. Then, for every acquisition (Rad) and group ( $U_{10}$ ), we sort the 25  $Q_{min}$  values from minimum to maximum as we can relate the plume position in this distribution to their associated POD. For instance, the  $3^{rd}$  plume of each distribution is related to a POD = 12 % (3/25), meaning that the  $Q_{min}$  from this plume is the Q value at which we can detect 12 % of the plumes at a given  $U_{10}$  and Rad values.

We group those plumes that exhibit the same POD value. For each set of plumes associated with a particular POD, we then relate  $U_{10}$  to  $Q_{min}$  using a quadratic fit, which allows us to capture the general relationship between the two variables. We acknowledge that increasing the number of plumes per group could provide greater constraint on the relationship between  $U_{10}$  and  $Q_{min}$ . However, the process of generating plumes is highly time-intensive and, as a result, our simulation dataset was limited in size. Fig. 4 shows an example for an EnMAP acquisition with  $Rad = 0.23 \ Wm^{-2}sr^{-1}\mu m^{-1}$ . It is then possible to relate Rad,  $U_{10}$  and  $Q_{min}$  for a given POD. We first attempted to predict Rad as a function of  $U_{10}$  and  $Q_{min}$ , using a simple quadratic relationship. We find a low  $R^2 = 0.5$ , as the exponential relationship between  $Q_{min}$  and Rad (Fig. 5 – right) does not match the polynomial nature of the quadratic function. Nevertheless, considering the strong relationship between Rad

and  $\sigma_{\Delta XCH_4}$  (see Section 3.1), we can use  $\sigma_{\Delta XCH_4}$  as a proxy of Rad. Due to the more suitable relationship between the  $\sigma_{\Delta XCH_4}$  and  $Q_{min}$  (Fig. 5 - left), we obtain a better fit of  $R^2$  = 0.97. Therefore, the quadratic function to fit can be expressed as

$$\sigma_{\Delta XCH_4}(\mathbf{U}_{10}, \mathbf{Q}_{\min}) = \underline{\mathbf{a}}c + \underline{\mathbf{b}}\underline{\mathbf{d}}\mathbf{U}_{10} + \underline{\mathbf{c}}e\mathbf{Q}_{\min} + \underline{\mathbf{d}}f\mathbf{U}_{10}\mathbf{Q}_{\min} + \underline{\mathbf{e}}g\mathbf{U}_{10}^2 + \underline{f}\underline{\mathbf{b}}\mathbf{Q}_{\min}^2$$

$$\tag{6}$$

where  $\frac{a-f}{c-h}$  are parameters to fit.

We acknowledge that increasing the number of plumes per group and the number of groups could provide greater constraint on the relationship between  $U_{10}$  and  $Q_{min}$ . However, the process of generating plumes is highly time-intensive and, as a result, our simulation dataset was limited in size. According to Ouerghi et al. (2025), a time difference of 120 s between adjacent plume instances leads to considerably different plume concentration distributions, while 30 s only leads to some minor changes. Due to the limited size of our simulated dataset, we find that the most optimal trade-off to reduce correlation between adjacent plumes is to set a minimum time difference of 60 s. At this value, changes in the distribution are more notable than in the 30 s case, but some correlation between plumes still exists. We will take this into account in Section 4.

**Table 1.** Mean and 1-standard deviation of the  $U_{10}$  distribution related to the different simulation groups (G) containing synthethic methane plumes.

|                                             | G1                           | G2    | G3    | G4    | G5    | G6   | G7 G8 G9 G10 G11                        |
|---------------------------------------------|------------------------------|-------|-------|-------|-------|------|-----------------------------------------|
| $mean(U_{10})$ $(m/s)$                      | <del>1.618 1.739</del> 2.145 | 2.286 | 3.11  | 3.329 | 3.747 | 4.15 | <del>4.533</del> 4.72 <del>5.237</del>  |
| $\operatorname{std}(\mathbf{U}_{10})$ (m/s) | <del>0.039 0.02</del> 0.029  | 0.035 | 0.034 | 0.079 | 0.057 | 0.1  | <del>0.02</del> -0.045 <del>0.072</del> |

We test this model using real emissions (Table B1), where plumes have been identified under specific  $\sigma_{\Delta XCH_4}$  and  $U_{10}$  values. We compare the Q value of the plume with the Q values related to different POD at these very same conditions. In the case of PRISMA, we use the EnMAP model due to their similarity, although EnMAP is expected to retrieve lower detection limits due to its higher sensitivity to methane (Roger et al., 2024).

## 2.4 Study sites

We collect multiple acquisitions from the EnMAP and EMIT satellite missions capturing offshore areas distributed around the world (Table C1 and C2), covering a representative range of *Rad* levels in order to apply the methodologies described in Section 2.2 and 2.3. In addition, we also gathered acquisitions from EnMAP, EMIT, and also PRISMA where at least one emission has been detected (Table B1). The acquisitions were obtained from the archive located in the EnMAP data portal (DLR, 2024), the PRISMA data portal (ASI, 2024), and the EarthData portal (NASA, 2024) for EnMAP, PRISMA, and EMIT, respectively. In the top panel from Fig. 6, we can see the location of the acquisitions from EnMAP (green dots) and EMIT (red dots) shown in Table C1 and C2, and the locations of the detected offshore emissions with EnMAP, PRISMA, and EMIT data listed in Table B1 (black triangles). Moreover, in the bottom panel we can observe some examples of detected methane plumes

**Figure 4.** Relationship between  $U_{10}$  vs  $Q_{min}$  obtained from WRF-LES simulations integrated in an EnMAP L1 acquisition with a level of radiance =  $0.23~Wm^{-2}sr^{-1}\mu m^{-1}$  at 2131 nm (points) and the related quadratic fit curves (dashed lines) that capture the general tendency for POD = 12~% (red), 52~% (blue), and 92~% (green).

Figure 5. Relationship between  $\sigma_{\Delta XCH_4}$  and  $Q_{min}$  (left) and Rad vs  $Q_{min}$  (right) for the EnMAP (blue) and EMIT (red) acquisitions where WRF-LES simulations have been integrated.

(pointed with red arrows) with EMIT data in the Mexican GoM, with PRISMA data in the Gulf of Guinea, and with EnMAP data in the Caspian Sea.

**Figure 6.** In top panel, locations from the EnMAP (green dots) and EMIT (red dots) acquisitions of Table C1 and C2 and from the methane plumes (black triangles) detected using EnMAP, PRISMA, and EMIT data of Table B1. In the bottom panel, examples of detected plumes overlaid on radiance maps are shown using data from EnMAP (framed in green), EMIT (framed in red), and PRISMA (framed in blue).

#### 3 Results

### 3.1 Analysis of potential factors affecting methane emission detection in offshore areas

In Fig. 7, we show the relationship between Rad and several parameters for every acquisition listed in Table C1 and C2, obtained using the EnMAP (left) and EMIT (right) instruments, respectively. There is a clear exponential trend between Rad and  $\sigma_{\Delta XCH_4}$  (green) that drives methane retrieval precision. Acquisitions with higher Rad values benefit from a better retrieval precision, which improves the emission detection performance. At the same Rad levels, EMIT outperforms the retrieval precision in comparison to that of EnMAP due to a higher SNR (blue), which can be attributed to the lower spatial resolution.

For both missions, we observe that higher Rad levels mostly occur at lower SGA (red) values, where there is a closer-to-sun glint configuration at the time of acquisition. On the contrary, a low SGA is not always equivalent to high Rad levels since this parameter is considered under the assumption of a flat water surface. Therefore, it does not account for variations in the local incidence angle of water caused by wind-induced waves. It is also important to note that greater surface roughness in water (denoted as  $\sigma_{Rad}/Rad$ ) results in a more extended sun glint reflection (Capderou, 2014). This concept is illustrated with a polar plot in Fig. 8, which shows a mock example of the SGA values where methane emission detection with the EnMAP instrument is feasible for a given plume. It assumes SZA =  $20^{\circ}$  and considers cases with low (red area) and high (orange area) water roughness. In this plot, the radial and angular coordinates correspond to zenithal and relative azimuth angles. We can observe that the detection area expands as water roughness increases from low to high, resulting in detections feasible across a broader range of geometric configurations.

The left panel from Fig. 9 shows a histogram of the SGA values from the EMIT acquisitions listed in Table C2, where the most frequent values are located in the 10°-15° range. Although these points exhibit similar SGA values, they present different *Rad* levels. In the right panel of Fig. 9, a positive correlation between *Rad* and σ<sub>Rad</sub>/Rad is observed, which we attempt to fit with a linear function. We also show the normalized instrument noise from the EMIT instrument, i.e. NEdL/Rad, to illustrate that the σ<sub>Rad</sub>/Rad trend cannot be explained by this magnitude. Therefore and therefore is mostly driven by surface roughness.

Then, higher surface roughness will generally lead to greater radiance levels for similar SGA values. It is important to note that offshore acquisitions mainly consist of water pixels, implying spectral homogeneity across the entire scene. Then, differences in the radiance levels among pixels caused by water roughness will be the most important contribution to surface heterogeneity. Our results suggest that this contribution is not significant enough to impair detection. However, this is not always the case in land scenes, where surface artifacts with different spectral shapes are more common.

Due to the relatively low VZA values from the EnMAP and EMIT instruments, there is a positive correlation between SZA and SGA (see Eq. 3), which generally leads to closer-to-sun glint acquisitions at lower SZA. Moreover, at the same SZA, EnMAP exhibits more flexibility to achieve this configuration as it has the ability to point in the across-track direction. A more detailed analysis can be found in Appendix D.

The IA parameter is considered an important magnitude to acquire high levels of radiance when using GHGSat data in offshore areas (MacLean et al., 2024). Due to the superior pointing ability of GHGSat, these instruments can get acquisitions

Figure 7. Scatter plots of Rad against SGA (red), SNR at the ~2131 nm band (blue), and  $\sigma_{\Delta XCH_4}$  (green) for the EnMAP (left) and EMIT (right) acquisitions showed in Table C1 and C2.

with high VZA values, which results in substantially higher IA values in comparison to those of EnMAP and EMIT. Then, this parameter does not play an important role for these two instruments, as detailed in Appendix E.

It is worth mentioning that the EnMAP mission follows a sun-synchronous orbit, ensuring that the instrument always captures data from the same location at a consistent local time. However, data related to different locations is acquired at different local times. According to our data, local times from EnMAP acquisitions were approximately constrained between 10:00 and 14:00. Thus, there is a maximum difference of 2 hours to the noon, where SZA gets to its minimum value. On-board the International Space Station (ISS), the EMIT mission follows a non-fixed orbit, resulting in different acquisition local times for each data take and location. Regarding the EMIT data from this study, we obtained acquisitions captured roughly between 10:00 and 18:00 local time that leads to a higher maximum difference of 6 hours to the noon. Nevertheless, in this context, the flexibility of the EMIT orbit allows for more favorable acquisitions with sun glint, as it is not restricted by the fixed local time of data takes like EnMAP. However, the EMIT orbit is unpredictable, making these more favorable acquisitions difficult to anticipate, while EnMAP enables users to schedule data acquisitions with precise angular configurations.

#### 3.2 Probability of detection of methane plumes in offshore areas

In Fig. 10, we show the Q values in which an emission can be detected with EnMAP (top row) and EMIT (bottom row) regarding a given  $\sigma_{\Delta XCH_4}$  and U<sub>10</sub> values for a POD of 12 % ( $\sim$  10%) (left), 52 % ( $\sim$  50%) (center), and 92 % ( $\sim$  90%)

**Figure 8.** Polar plot showing a mock example of the SGA values where methane emission detection with the EnMAP instrument is feasible for a methane emission, assuming  $SZA = 20^{\circ}$ . The radial and angular coordinates correspond to zenithal and relative azimuth angles. Combinations of geometric configurations where the plume can be detected are indicated by red and orange areas for low and high water roughness cases, respectively.

Figure 9. Histogram showing the SGA distribution from the EMIT acquisitions from Table C2 (left) and the scatter plot of Rad against  $\sigma_{Rad}/Rad$  from those EnMAP points with SGA values ranging from 10° to 15° (right), where a higher density of data is located. A normalized NEdL curve (black) and a linear fit to the data points (red) are also illustrated.

(right). To these plots, we overlay the Q values extracted from the POD models using the  $\sigma_{\Delta XCH_4}$  and  $U_{10}$  values related to the images listed in Table B1. These are PRISMA (crosses), EnMAP (points), and EMIT (squares) acquisitions in which we detected real emissions. Note that only those acquisitions in which their related  $U_{10}$  values are within the wind speed range

from the simulation groups in Table 1 are included. We observe that for a given  $U_{10}$  and  $\sigma_{\Delta XCH_4}$  values, the Q value increases for higher PODs. This exhibits the consistency of the models, where higher Q values are needed for greater PODs. On the other hand, we observe that regarding the same  $U_{10}$  and Q values for both instrument,  $\sigma_{\Delta XCH_4}$  is higher for EnMAP than for EMIT. This implies that EnMAP requires a less demanding retrieval precision for detection, which can be explained with its better spatial resolution. Note that, regarding the lower sensitivity to methane from PRISMA as compared to EnMAP (Roger et al., 2024), the former will probably require a more demanding retrieval precision than the latter. Moreover, the lower spatial resolution of EMIT and the higher area needed to consider a plume detectable due to the selection of N=10 (see Section 2.1) are factors that make detection more challenging. However, it is important to note that EMIT typically exhibits better retrieval precision due to higher SNR.

Figure 10.  $\sigma_{\Delta XCH_4}$  dependency with Q and  $U_{10}$  extracted from EnMAP (top row) and EMIT (bottom row) data for (from left to right) POD  $\sim 10 \%$ ,  $\sim 50 \%$ , and  $\sim 90 \%$ . EnMAP (points), PRISMA (crosses), and EMIT (squares) detections are overlaid with a Q value associated to the retrieval precision and  $U_{10}$  from the acquisition.

We aim to compare the detection capability between instruments using Rad instead of  $\sigma_{\Delta XCH_4}$ . Then, we can assess both sensors under the same input signal, providing a more consistent basis for comparison. For this purpose, we fit  $\sigma_{\Delta XCH_4}$  to the associated Rad values from the acquisitions in Table C1 and C2 and obtain the dependency of Rad with the Q and  $U_{10}$  parameters (see Appendix F). As an output of this calculation, we will obtain a minimum Rad value for detection. Therefore, a sensor with a lower minimum Rad will provide a more favorable context for detection. Maps representing the difference between the EnMAP and EMIT minimum Rad ( $\Delta Rad = Rad_{EnMAP} - Rad_{EMIT}$ ) are shown in Fig. 11 for POD  $\sim$  10 % (left), 50 % (center), and 90 % (right) to see which instrument requires lower radiance levels for detection.  $\Delta Rad > 0$  indicates that EMIT is better suited for detection, while  $\Delta Rad < 0$  suggests that EnMAP is more favorable for this purpose. Dashed black

lines separating zones with negative and positives values of  $\Delta Rad$  are overlaid. The approximate Q and U<sub>10</sub> combinations that result in negative and positive  $\Delta Rad$  values can be expressed as

$$\Delta \operatorname{Rad}(Q, U_{10}) = \begin{cases} Q < A, \forall U_{10} & \to \Delta \operatorname{Rad} > 0\\ Q \ge A, \forall U_{10} & \to \Delta \operatorname{Rad} < 0 \end{cases}$$

$$\tag{7}$$

where a = 1.5 t/h, b = 4 m/s, c

$$\Delta \text{Rad}(Q, U_{10}) = \begin{cases} Q < B, U_{10} \ge C & \rightarrow \Delta \text{Rad} > 0 \\ Q < B, U_{10} < C & \rightarrow \Delta \text{Rad} < 0 \\ B < Q < D, \forall U_{10} & \rightarrow \Delta \text{Rad} < 0 \\ Q \ge D, \forall U_{10} & \rightarrow \Delta \text{Rad} > 0 \end{cases}$$

$$(8)$$

where POD  $\sim$  10 % follows Eq. 7 with A = 5 t/h for; POD  $\sim$  10 %; a-50 % follows Eq. 8 with  $B = \frac{2}{3}$  t/h,  $\frac{b}{b}C = \frac{4}{4}$ .1 m/s,  $\frac{c}{a}$  and  $\frac{d}{d}D = \frac{7}{4}$  t/h,  $\frac{b}{b}C = \frac{4}{5}$  m/s,  $\frac{c}{a}$  and  $\frac{d}{d}D = \frac{9}{4}$  t/h,  $\frac{b}{b}C = \frac{4}{5}$  m/s,  $\frac{c}{a}$  and  $\frac{d}{d}D = \frac{9}{4}$  t/h,  $\frac{b}{b}C = \frac{4}{5}$  m/s,  $\frac{c}{a}$  and  $\frac{d}{d}D = \frac{9}{4}$  t/h,  $\frac{b}{b}C = \frac{4}{5}$  m/s,  $\frac{c}{a}$  and  $\frac{d}{d}D = \frac{9}{4}$  t/h,  $\frac{b}{b}C = \frac{4}{5}$  m/s,  $\frac{c}{a}$  and  $\frac{d}{d}D = \frac{9}{4}$  t/h,  $\frac{b}{b}C = \frac{4}{5}$  m/s,  $\frac{c}{a}$  and  $\frac{d}{d}D = \frac{9}{4}$  t/h,  $\frac{b}{b}C = \frac{4}{5}$  m/s,  $\frac{c}{a}$  and  $\frac{d}{d}D = \frac{9}{4}$  t/h,  $\frac{b}{b}C = \frac{4}{5}$  m/s,  $\frac{c}{a}$  and  $\frac{d}{d}D = \frac{9}{4}$  t/h,  $\frac{b}{d}C = \frac{4}{5}$  m/s,  $\frac{c}{d}$  and  $\frac{d}{d}D = \frac{9}{4}$  t/h,  $\frac{b}{d}C = \frac{4}{5}$  m/s,  $\frac{c}{d}$  and  $\frac{d}{d}D = \frac{9}{4}$  t/h,  $\frac{b}{d}C = \frac{4}{5}$  m/s,  $\frac{c}{d}$  and  $\frac{d}{d}D = \frac{9}{4}$  t/h,  $\frac{d}{d}D = \frac{9}{$ 

Figure 11. The difference between the minimum Rad values for detection from EnMAP and EMIT extracted from the models used for POD  $\sim 10\%$  (left), 50% (center), and 90% (right). Dashed black lines separating positive (detection more favorable for EMIT) and negative (detection more favorable for EnMAP) values of this parameter are overlaid.

Most combinations with Q < e lower than a certain threshold (A for POD  $\sim$  10 %, and D for POD  $\sim$  50 % and POD  $\sim$  90 %) show  $\Delta Rad < 0$  values, which indicates that EnMAP acquisitions will require a lower Rad for detection. However, for  $U_{10} > bC$  m/s and Q < eB t/h in the POD  $\sim$  50 % and POD  $\sim$  90 % cases, we find  $\Delta Rad > 0$  values. At high wind speeds, plumes are extended over larger areas with weaker concentrations, and EMIT seems to offer more favorable conditions for detection due to a very localized balance at these PODs. Nevertheless, at this high  $U_{10}$  range, stronger emissions meeting eB < Q < eD are more easily detected with EnMAP ( $\Delta Rad < 0$ ), which outperforms the EMIT balance due to a higher spatial resolution.

On the other hand, for approximately  $Q \ge e^*D$  emissions are so intense and widespread that the spatial resolution is not an advantage anymore, resulting in a better detection limit of EMIT (ΔRad > 0). The difference maps in Fig. 11 indicate that the EMIT local balance does not exist in POD ~ 10 %, but appears in POD ~ 50 % and shifts toward higher Q and U10 values as the POD increases. Similarly, the other transition from negative to positive ΔRad values occurs at higher Q values as POD increases, which highlights the importance of the better spatial resolution of EnMAP when considering most plumes.

On the other hand, since we have not extracted POD models for PRISMA, we cannot include it in the comparison. However, a performance similar to that of EnMAP is expected. Note that, for simplification, the joint correlation among wind speed, surface roughness, and radiance has not been explicitly considered in the POD models. Nevertheless, a relatively large number of samples under different combinations of these parameters have been used, providing some degree of representativeness of this correlation. Even so, a more intensive Rad sampling would further enhance the robustness of the models.

Using the POD  $\sim 10$  % model, we obtained the related flux rate values assuming the  $\sigma_{\Delta XCH_4}$  and  $U_{10}$  values from the images listed in Table B1, where real emissions were detected. According to our model, these flux rates represent the values at which we can only detect 10 % of the possible plumes, while the other 90 % would remain undetected. We compared these values to the estimated plume flux rates and found that most estimations were higher than the model values. Since most plumes would not be detected at the model values, the fact that the majority of our plumes exhibit higher flux rates supports the consistency of our model. However, there are 1 EnMAP and 2 PRISMA detections in which the estimations are lower than those from the model. After examination, we find that these detections do not pass the detection algorithm test. In the EnMAP case, the emission is too thin and the median filter from the algorithm removes it, while in the PRISMA cases the emission enhancement values are at retrieval background level and both plumes were identified as emissions only under a careful visual inspection. Although a threshold of  $1\sigma_{\Delta XCH_4}$  would be closer to the criteria used in visual inspection (Guanter et al., 2021), a threshold of  $2\sigma_{\Delta XCH_4}$  was set in the detection algorithm (see Section 2.1) to better separate background pixels from those related to methane emissions.

Due to the low temporal resolution of the PRISMA, EnMAP, and EMIT imaging spectrometers, a joint use of these missions will increase the probability to detect point source emissions in an area with potential emitters. Once data have been acquired, using radiance levels from the  $\sim 2131$  nm bands and the wind speed values from products such as GEOS-FP, will provide the flux rate related to the POD models (see Appendix F). This will allow to keep or discard images for plume detection after setting a criteria provided by the user, such as a flux rate threshold. Similarly, if we have the methane enhancement concentration retrievals, we could repeat the same exercise but using the results from Fig. 10.

## 4 Summary and conclusions

In this work, we collected EnMAP and EMIT acquisitions (around 70 each) capturing offshore areas and covering a representative range of radiance levels. We assessed the main parameters that have an impact on methane emission detection using these samples. Simulated plumes were integrated into real data to obtain real-like plumes, which allowed us to extract models

reproducing the detection limit conditions at different probability of detection values. Finally, we intercompared the detection capabilities from the EnMAP and EMIT instruments and used real emission detections to assess the models.

The typical low reflectance of water leads to noisy retrievals where detection of methane emissions is challenging. However, acquisitions taken at a mirror-like configuration benefit from the increase of radiance levels from the sun glint effect. The proximity to this particular configuration under the assumption of a flat surface is measured by the scattering glint angle. Lower values of this parameter result in acquisitions closer to the sun glint effect. Our results show that higher radiance levels were found at lower scattering glint angles. Moreover, this parameter is highly correlated to the solar zenith angle parameter. For the same solar zenith angle values, EnMAP can get closer to the sun glint angular configuration than EMIT due to a wider pointing range, although the specific sensor configuration needs to be tasked. Additionally, surface roughness shows a positive correlation with radiance, while the incidence angle has a negligible effect due to the low Fresnel coefficient values.

The detection limits from both instruments are assessed with the extracted POD models from this study, using WRF-LES simulations of plumes with a related  $U_{10}$  between 1.6-2.1 m/s and 5.2-4.7 m/s. We note that these models could be improved through expanded simulation of plumes with a greater diversity of shapes and a wider range of wind speeds. Nonetheless, we find that the higher spatial resolution of EnMAP generally leads to a lower required retrieval precision to detect a plume at a given  $U_{10}$  and Q as compared to EMIT. Due to the higher sensitivity to methane from EnMAP, PRISMA will have more demanding precision requirements. However, due to the superior SNR from EMIT coming from a lower spatial resolution, this instrument generally exhibits better retrieval precision than EnMAP. Moreover, we find that EnMAP requires lower radiance levels for detection at approximately Q 

Figure A1. Radiance image of the Baikal Lake acquired with the EnMAP instrument at the  $\sim 2131$  nm band on 26/09/2022, with a center latitude and longitude coordinates of  $55.4266^{\circ}$  and  $109.4987^{\circ}$ , respectively. The subset of  $500 \times 500$  pixels where calculations were applied is framed in red.

On the other hand, when downsampling, we might be including other noise sources such as surface variability. These are added to the pixel noise, but are independent from the instrument noise contribution. It is important to note this difference, since reduction of instrument noise is critical to obtain a higher SNR. It is also important to note that this is a simple approach to better understand instrument noise reduction and a more thorough analysis would be needed to obtain higher accuracy in our estimations. However, the analysis is out of the scope of this study.

## Appendix B: Methane emissions detected in offshore areas with the EnMAP, EMIT, and PRISMA instruments

**Table B1:** List of methane plumes detected in offshore areas with *EnMAP*, *PRISMA*, and *EMIT* data. *IMEO* and *JPL* sources refer to the UNEP's IMEO Methane Data portal and the NASA's JPL portal, respectively. The date information is in DD/M-M/YYYY format. Latitude and longitude represent the plume coordinates.

| Mission  | Site             | Date       | Latitude | Longitude | Q     | err(Q) | $\mathbf{U}_{10}$ | $\sigma_{\Delta XCH_4}$ | Source         |
|----------|------------------|------------|----------|-----------|-------|--------|-------------------|-------------------------|----------------|
| WHISSIOH |                  | Date       | (°)      | (°)       | (t/h) | (t/h)  | (m/s)             | (ppm)                   | Source         |
| EnMAP    | US GoM           | 01/07/2022 | 29.1110  | -90.4688  | 1.1   | 0.4    | 3.32              | 0.09                    | Roger, 2024b   |
| EnMAP    | US GoM           | 01/07/2022 | 29.1132  | -90.4828  | 0.6   | 0.2    | 3.32              | 0.09                    | Roger, 2024b   |
| EnMAP    | Mexican GoM      | 11/02/2024 | 19.5843  | -92.2324  | 70    | 20     | 7.05              | 0.202                   | Research       |
| EnMAP    | Caspian sea      | 12/07/2024 | 40.2833  | 50.7428   | 2.2   | 0.8    | 4.38              | 0.109                   | IMEO           |
| EnMAP    | Caspian sea      | 12/07/2024 | 40.2784  | 50.7533   | 2.6   | 0.9    | 4.38              | 0.109                   | IMEO           |
| EnMAP    | Caspian sea      | 12/07/2024 | 40.2153  | 50.9004   | 2.4   | 0.9    | 4.38              | 0.109                   | IMEO           |
| EnMAP    | Caspian sea      | 12/07/2024 | 40.2278  | 50.9106   | 2.5   | 0.9    | 4.38              | 0.109                   | IMEO           |
| EnMAP    | Caspian sea      | 12/07/2024 | 40.2344  | 50.9167   | 4     | 1      | 4.38              | 0.109                   | IMEO           |
| EnMAP    | Caspian sea      | 12/07/2024 | 40.2297  | 50.9242   | 4     | 1      | 4.38              | 0.109                   | IMEO           |
| PRISMA   | Mexican GoM      | 09/02/2024 | 19.5843  | -92.2324  | 90    | 20     | 9.68              | 0.417                   | IMEO           |
| PRISMA   | Gulf of Thailand | 24/04/2023 | 7.5941   | 102.9879  | 7     | 3      | 3.30              | 0.306                   | Valverde, 2023 |
| PRISMA   | Persian Gulf     | 14/08/2023 | 26.5878  | 52.0422   | 4     | 1      | 2.37              | 0.128                   | IMEO           |
| PRISMA   | Gulf of Guinea   | 20/09/2023 | -5.6371  | 11.8510   | 1.9   | 0.7    | 3.93              | 0.175                   | IMEO           |
| PRISMA   | Gulf of Guinea   | 22/11/2022 | -6.9876  | 12.3705   | 5     | 1      | 1.77              | 0.364                   | IMEO           |
| PRISMA   | Gulf of Guinea   | 18/03/2023 | -7.0973  | 12.3345   | 1.9   | 0.7    | 2.66              | 0.371                   | IMEO           |
| EMIT     | Gulf of Guinea   | 18/02/2024 | -7.1714  | 12.3865   | 5     | 1      | 4.32              | 0.165                   | IMEO           |
| EMIT     | Persian Gulf     | 02/08/2023 | 29.6310  | 48.8638   | 1.9   | 0.7    | 3.46              | 0.046                   | JPL            |
| EMIT     | Persian Gulf     | 02/08/2023 | 29.6384  | 48.8192   | 3     | 1      | 3.46              | 0.046                   | JPL            |
| EMIT     | Persian Gulf     | 02/08/2023 | 29.6486  | 48.8509   | 1.6   | 0.6    | 3.46              | 0.046                   | JPL            |
| EMIT     | Mexican GoM      | 21/04/2024 | 19.5843  | -92.2324  | 22    | 7      | 4.66              | 0.046                   | IMEO           |
| EMIT     | Mexican GoM      | 24/12/2023 | 19.5633  | -92.2350  | 10    | 2      | 8.17              | 0.114                   | Research       |
| EMIT     | Persian Gulf     | 25/07/2024 | 48.7984  | 29.7089   | 1.6   | 0.4    | 1.48              | 0.077                   | IMEO           |
| EMIT     | Persian Gulf     | 25/07/2024 | 48.8144  | 29.6810   | 3.2   | 0.9    | 1.48              | 0.077                   | IMEO           |
| EMIT     | Persian Gulf     | 25/07/2024 | 48.8599  | 29.6418   | 2.7   | 0.7    | 1.48              | 0.077                   | IMEO           |

Appendix C: EnMAP and EMIT acquisitions with a representative range of Rad values

**Table C1:** List of the collected EnMAP acquisitions capturing offshore areas. The date information is in DD/MM/YYYY format. *xi*, *yi*, *xe*, *ye* are the pixel coordinates describing the initial (*i*) and final (*e*) rows (*x*) and columns (*y*) from the selected subsets of the scene. The latitude and longitude indicate the center coordinates of the entire acquisition and therefore do not match the subset coordinates.

|         |                   |            | Acquisition  | Acquisition   | Subset                |
|---------|-------------------|------------|--------------|---------------|-----------------------|
| Mission | Location          | Date       | Central      | Central       | Pixel Coordinates     |
|         |                   |            | Latitude (°) | Longitude (°) | (xi, yi, xe, ye)      |
| EnMAP   | Red Sea           | 26/10/2022 | 12.7296      | 43.4464       | (300, 150, 350, 650)  |
| EnMAP   | Red Sea           | 03/05/2023 | 13.1111      | 42.9384       | (300, 0, 350, 500)    |
| EnMAP   | US GoM            | 01/07/2022 | 29.0234      | -90.3719      | (470, 500, 520, 1000) |
| EnMAP   | Gulf of Guinea    | 31/07/2022 | 0.4465       | 6.6446        | (150, 0, 200, 500)    |
| EnMAP   | Persian Gulf      | 21/04/2023 | 26.7957      | 51.9479       | (500, 500, 550, 1000) |
| EnMAP   | Coral Sea         | 27/03/2023 | -13.2453     | 167.5125      | (141, 111, 191, 611)  |
| EnMAP   | Pacific Ocean     | 03/08/2022 | 20.5998      | -157.0775     | (150, 460, 200, 960)  |
| EnMAP   | Pacific Ocean     | 03/08/2022 | 20.5998      | -157.0775     | (840, 460, 890, 960)  |
| EnMAP   | Pacific Ocean     | 08/04/2023 | 1.0163       | -143.5149     | (700, 500, 750, 1000) |
| EnMAP   | Gulf of Guinea    | 27/07/2023 | -5.4954      | 11.8266       | (500, 10, 550, 510)   |
| EnMAP   | Gulf of Guinea    | 27/07/2023 | -5.2228      | 11.8843       | (700, 10, 750, 510)   |
| EnMAP   | Persian Gulf      | 21/04/2023 | 26.7957      | 51.9479       | (540, 10, 590, 510)   |
| EnMAP   | Mediterranean Sea | 27/03/2023 | 35.5254      | 12.4765       | (480, 10, 530, 510)   |
| EnMAP   | Gulf of Guinea    | 31/07/2022 | 0.4465       | 6.6446        | (100, 10, 150, 510)   |
| EnMAP   | Bohai Sea         | 22/06/2023 | 38.6003      | 118.824       | (270, 50, 320, 550)   |
| EnMAP   | Bohai Sea         | 16/07/2023 | 38.7741      | 118.9396      | (250, 1, 300, 501)    |
| EnMAP   | Bohai Sea         | 16/07/2023 | 38.505       | 118.8535      | (105, 307, 155, 807)  |
| EnMAP   | Bohai Sea         | 16/07/2023 | 38.2356      | 118.7683      | (430, 1, 480, 501)    |
| EnMAP   | Bohai Sea         | 27/07/2023 | 38.3953      | 118.7892      | (733, 1, 783, 501)    |
| EnMAP   | Bohai Sea         | 27/07/2023 | 38.6664      | 118.8608      | (736, 180, 786, 680)  |
| EnMAP   | Bohai Sea         | 27/07/2023 | 38.1239      | 118.7182      | (96, 80, 146, 580)    |
| EnMAP   | US GoM            | 04/04/2023 | 28.7662      | -90.8399      | (790, 244, 840, 744)  |
| EnMAP   | Atlantic Ocean    | 31/03/2023 | 14.9351      | -17.3708      | (538, 50, 588, 550)   |
| EnMAP   | Atlantic Ocean    | 11/01/2024 | 14.7862      | -17.3172      | (80, 70, 130, 570)    |
| EnMAP   | Atlantic Ocean    | 07/02/2024 | 14.8008      | -17.4861      | (560, 5, 610, 505)    |
| EnMAP   | South China Sea   | 16/02/2024 | 6.2206       | 116.1832      | (100, 1, 150, 501)    |

| EnMAP | Persian Gulf    | 29/03/2024 | 28.1033  | 48.9985  | (500, 50, 550, 550)   |
|-------|-----------------|------------|----------|----------|-----------------------|
| EnMAP | Persian Gulf    | 24/01/2024 | 29.3781  | 48.7502  | (20, 330, 70, 830)    |
| EnMAP | South China Sea | 20/02/2024 | 6.2405   | 116.191  | (15, 100, 65, 600)    |
| EnMAP | Persian Gulf    | 10/03/2024 | 29.4755  | 48.7779  | (570, 90, 620, 590)   |
| EnMAP | Persian Gulf    | 29/03/2024 | 28.3752  | 49.062   | (484, 491, 534, 991)  |
| EnMAP | Bay of Bengal   | 27/03/2023 | 13.5028  | 80.3333  | (685, 200, 735, 700)  |
| EnMAP | Bay of Bengal   | 12/03/2024 | 9.671    | 81.207   | (690, 200, 740, 700)  |
| EnMAP | Bay of Bengal   | 27/03/2023 | 12.4116  | 80.104   | (838, 136, 888, 636)  |
| EnMAP | Bay of Bengal   | 27/03/2023 | 13.7757  | 80.3907  | (911, 5, 961, 505)    |
| EnMAP | Bay of Bengal   | 01/05/2024 | 9.6559   | 81.0998  | (750, 120, 800, 620)  |
| EnMAP | Bay of Bengal   | 27/03/2023 | 12.9572  | 80.2185  | (850, 30, 900, 530)   |
| EnMAP | Bay of Bengal   | 27/03/2023 | 13.2302  | 80.2759  | (820, 500, 870, 1000) |
| EnMAP | Atlantic Ocean  | 12/04/2024 | 40.579   | -8.7261  | (100, 420, 150, 920)  |
| EnMAP | Atlantic Ocean  | 12/04/2024 | 40.8476  | -8.636   | (10, 10, 60, 510)     |
| EnMAP | Atlantic Ocean  | 12/04/2024 | 40.3102  | -8.8157  | (50, 50, 100, 550)    |
| EnMAP | Atlantic Ocean  | 12/04/2024 | 40.0414  | -8.9047  | (60, 10, 110, 510)    |
| EnMAP | Atlantic Ocean  | 12/04/2024 | 39.7728  | -8.9934  | (100, 100, 150, 600)  |
| EnMAP | Atlantic Ocean  | 12/04/2024 | 39.5039  | -9.0812  | (70, 60, 120, 560)    |
| EnMAP | Bohai Sea       | 22/06/2023 | 38.3292  | 118.7591 | (470, 0, 520, 500)    |
| EnMAP | Tasman Sea      | 28/02/2024 | -33.3215 | 151.7668 | (700, 50, 750, 550)   |
| EnMAP | Tasman Sea      | 28/02/2024 | -33.0524 | 151.8451 | (900, 80, 950, 580)   |
| EnMAP | Bay of Biscay   | 10/02/2024 | 46.8929  | -2.151   | (380, 400, 430, 900)  |
| EnMAP | Bay of Biscay   | 03/02/2024 | 43.4593  | -8.2364  | (95, 25, 145, 525)    |
| EnMAP | Bay of Biscay   | 12/04/2024 | 43.7982  | -7.6008  | (550, 10, 600, 510)   |
| EnMAP | Atlantic Ocean  | 07/01/2024 | 20.6417  | -16.6475 | (100, 250, 150, 750)  |
| EnMAP | Atlantic Ocean  | 11/01/2024 | 15.0576  | -17.2539 | (380, 200, 430, 700)  |
| EnMAP | Persian Gulf    | 18/09/2023 | 26.8925  | 52.1353  | (125, 100, 175, 600)  |
| EnMAP | Persian Gulf    | 08/01/2024 | 26.7363  | 52.0378  | (180, 15, 230, 515)   |
| EnMAP | Persian Gulf    | 18/09/2023 | 26.6217  | 52.0665  | (870, 5, 920, 505)    |
| EnMAP | Persian Gulf    | 25/11/2022 | 26.0146  | 51.6557  | (270, 5, 320, 505)    |
| EnMAP | Persian Gulf    | 08/01/2024 | 26.4657  | 51.967   | (710, 60, 760, 560)   |
| EnMAP | Persian Gulf    | 18/09/2023 | 26.0795  | 51.9299  | (540, 120, 590, 620)  |
| EnMAP | Persian Gulf    | 18/09/2023 | 26.3505  | 51.9982  | (50, 299, 100, 799)   |

| EnMAP | Persian Gulf  | 21/04/2023 | 27.0658 | 52.023  | (360, 170, 410, 670) |
|-------|---------------|------------|---------|---------|----------------------|
| EnMAP | Persian Gulf  | 18/09/2023 | 27.1631 | 52.2045 | (100, 330, 150, 830) |
| EnMAP | Gulf of Cádiz | 20/09/2023 | 36.701  | -6.4914 | (380, 440, 430, 940) |
| EnMAP | Gulf of Cádiz | 01/05/2024 | 36.5772 | -6.3399 | (30, 50, 80, 550)    |
| EnMAP | Gulf of Cádiz | 01/05/2024 | 36.3091 | -6.4282 | (816, 144, 866, 644) |
| EnMAP | Gulf of Cádiz | 12/03/2024 | 36.575  | -6.3427 | (40, 50, 90, 550)    |
| EnMAP | Gulf of Cádiz | 24/07/2023 | 36.6352 | -6.461  | (160, 70, 210, 570)  |
| EnMAP | Gulf of Cádiz | 20/09/2023 | 36.4318 | -6.5736 | (100, 80, 150, 580)  |

**Table C2:** List of the collected EMIT acquisitions capturing offshore areas. The date information is in DD/MM/YYYY format. xi, yi, xe, ye are the pixel coordinates describing the initial (i) and final (e) rows (x) and columns (y) from the selected subsets of the scene. The latitude and longitude indicate the center coordinates of the entire acquisition and therefore do not match the subset coordinates.

|         |                 |            | Acquisition  | Acquisition   | Subset                  |
|---------|-----------------|------------|--------------|---------------|-------------------------|
| Mission | Location        | Date       | Central      | Central       | Pixel Coordinates       |
|         |                 |            | Latitude (°) | Longitude (°) | (xi, yi, xe, ye)        |
| EMIT    | South China Sea | 31/08/2023 | -0.0255      | 109.2158      | (10, 435, 60, 935)      |
| EMIT    | US GoM          | 22/04/2023 | 28.8138      | -90.2159      | (400, 1350, 450, 1850)  |
| EMIT    | US GoM          | 03/04/2023 | 28.6267      | -91.2574      | (250, 0, 300, 500)      |
| EMIT    | Persian Gulf    | 03/08/2023 | 26.1141      | 51.6975       | (900, 500, 950, 1000)   |
| EMIT    | Persian Gulf    | 27/05/2023 | 26.5021      | 53.1674       | (700, 0, 750, 500)      |
| EMIT    | Persian Gulf    | 23/08/2023 | 17.3096      | -23.013       | (950, 216, 1000, 716)   |
| EMIT    | Persian Gulf    | 23/08/2023 | 17.3096      | -23.013       | (1150, 216, 1200, 716)  |
| EMIT    | Persian Gulf    | 23/08/2023 | 17.3096      | -23.013       | (732, 705, 782, 1205)   |
| EMIT    | Persian Gulf    | 29/03/2024 | 17.3788      | -66.9807      | (250, 80, 300, 580)     |
| EMIT    | Persian Gulf    | 01/08/2023 | 37.7023      | 53.1165       | (250, 650, 300, 1150)   |
| EMIT    | Atlantic Ocean  | 27/04/2024 | -24.1512     | -44.8827      | (870, 550, 920, 1050)   |
| EMIT    | Atlantic Ocean  | 26/12/2023 | -24.8428     | -46.1521      | (80, 1450, 130, 1950)   |
| EMIT    | Bohai Sea       | 24/04/2024 | 37.5555      | 119.6206      | (800, 400, 850, 900)    |
| EMIT    | Bohai Sea       | 13/04/2024 | 39.2372      | 119.7329      | (40, 600, 90, 1100)     |
| EMIT    | Bohai Sea       | 24/04/2024 | 38.5047      | 118.1946      | (790, 25, 840, 525)     |
| EMIT    | Bohai Sea       | 24/04/2024 | 38.0327      | 118.9126      | (1100, 0, 1150, 500)    |
| EMIT    | Bohai Sea       | 29/03/2024 | 37.9423      | 119.8552      | (470, 110, 520, 610)    |
| EMIT    | Bohai Sea       | 23/02/2024 | 39.5042      | 119.3408      | (1120, 580, 1170, 1080) |

| EMIT | Atlantic Ocean     | 23/08/2023 | 16.5183  | -22.3824  | (1000, 40, 1050, 540)    |
|------|--------------------|------------|----------|-----------|--------------------------|
| EMIT | Atlantic Ocean     | 29/12/2023 | -14.0168 | -37.1166  | (350, 10, 400, 510)      |
| EMIT | South China Sea    | 26/04/2024 | 6.0105   | 114.7137  | (250, 200, 300, 700)     |
| EMIT | Caspian Sea        | 01/06/2023 | 39.3639  | 52.6134   | (470, 178, 520, 678)     |
| EMIT | Gulf of Tonkin     | 29/07/2023 | 18.1658  | 108.5886  | (100, 10, 150, 510)      |
| EMIT | Pacific Ocean      | 26/02/2024 | -14.7652 | -169.0183 | (180, 790, 230, 1290)    |
| EMIT | Pacific Ocean      | 22/01/2024 | -13.6707 | -168.1531 | (1000, 1500, 1050, 2000) |
| EMIT | Grau Sea           | 26/02/2024 | -5.362   | -81.5622  | (10, 50, 60, 550)        |
| EMIT | Grau Sea           | 26/02/2024 | -4.7607  | -81.9904  | (400, 500, 450, 1000)    |
| EMIT | Mexican GoM        | 23/06/2024 | 20.6947  | -93.1831  | (900, 720, 950, 1220)    |
| EMIT | US GoM             | 21/06/2024 | 28.5885  | -89.006   | (820, 1115, 870, 1615)   |
| EMIT | Gulf of Guinea     | 17/07/2024 | -14.3236 | 12.5345   | (1100, 0, 1150, 500)     |
| EMIT | South China Sea    | 25/06/2024 | 11.2774  | 115.1276  | (625, 390, 675, 890)     |
| EMIT | Pacific Ocean      | 23/06/2024 | 18.8196  | -115.2001 | (230, 70, 280, 570)      |
| EMIT | Pacific Ocean      | 23/06/2024 | 18.8196  | -115.2001 | (827, 0, 877, 500)       |
| EMIT | Pacific Ocean      | 23/06/2024 | 18.8196  | -115.2001 | (900, 0, 950, 500)       |
| EMIT | Atlantic Ocean     | 19/07/2024 | 30.9114  | -10.2256  | (300, 0, 350, 500)       |
| EMIT | Atlantic Ocean     | 19/07/2024 | 30.9114  | -10.2256  | (600, 0, 650, 500)       |
| EMIT | Atlantic Ocean     | 19/07/2024 | 30.9114  | -10.2256  | (750, 0, 800, 500)       |
| EMIT | Atlantic Ocean     | 19/07/2024 | 30.9114  | -10.2256  | (1100, 0, 1150, 500)     |
| EMIT | Atlantic Ocean     | 29/06/2024 | -4.8324  | -37.9213  | (1100, 0, 1150, 500)     |
| EMIT | Pacific Ocean      | 22/06/2024 | 7.4769   | 170.6305  | (0, 541, 50, 1041)       |
| EMIT | Pacific Ocean      | 22/06/2024 | 7.4769   | 170.6305  | (230, 630, 280, 1130)    |
| EMIT | Pacific Ocean      | 25/06/2024 | 7.6871   | 171.0004  | (850, 0, 900, 500)       |
| EMIT | Pacific Ocean      | 25/06/2024 | 7.6871   | 171.0004  | (730, 0, 780, 500)       |
| EMIT | Mozambique Channel | 20/05/2024 | -23.4568 | 43.8244   | (900, 0, 950, 500)       |
| EMIT | Mozambique Channel | 20/05/2024 | -23.4568 | 43.8244   | (1050, 0, 1100, 500)     |
| EMIT | Mediterranean Sea  | 24/05/2024 | 37.202   | 3.2676    | (0, 200, 50, 700)        |
| EMIT | Mediterranean Sea  | 24/05/2024 | 37.202   | 3.2676    | (300, 200, 350, 700)     |
| EMIT | Mediterranean Sea  | 24/05/2024 | 37.202   | 3.2676    | (600, 200, 650, 700)     |
| EMIT | Mediterranean Sea  | 22/06/2024 | 42.1705  | 16.3254   | (1100, 1600, 1150, 2100) |
| EMIT | Mediterranean Sea  | 22/06/2024 | 42.1705  | 16.3254   | (560, 1930, 610, 2430)   |
| EMIT | Mediterranean Sea  | 22/06/2024 | 42.1705  | 16.3254   | (250, 990, 300, 1490)    |

| EMIT | Mediterranean Sea | 22/06/2024 | 42.1705 | 16.3254 | (840, 0, 890, 500)       |
|------|-------------------|------------|---------|---------|--------------------------|
| EMIT | Mediterranean Sea | 22/06/2024 | 42.1705 | 16.3254 | (200, 1640, 250, 2140)   |
| EMIT | Mediterranean Sea | 22/06/2024 | 42.1705 | 16.3254 | (350, 150, 400, 650)     |
| EMIT | Mediterranean Sea | 09/06/2024 | 40.9381 | 10.3148 | (900, 620, 950, 1120)    |
| EMIT | Mediterranean Sea | 09/06/2024 | 40.9381 | 10.3148 | (150, 670, 200, 1170)    |
| EMIT | Mediterranean Sea | 09/06/2024 | 40.9381 | 10.3148 | (330, 670, 380, 1170)    |
| EMIT | Caspian Sea       | 07/06/2024 | 38.7006 | 48.6758 | (970, 750, 1020, 1250)   |
| EMIT | Caspian Sea       | 07/06/2024 | 38.7006 | 48.6758 | (1050, 750, 1100, 1250)  |
| EMIT | Black Sea         | 25/06/2024 | 41.6082 | 35.4566 | (1050, 0, 1100, 500)     |
| EMIT | Black Sea         | 25/06/2024 | 41.6082 | 35.4566 | (830, 0, 880, 500)       |
| EMIT | Black Sea         | 22/06/2024 | 45.0752 | 33.6723 | (0, 0, 50, 500)          |
| EMIT | Black Sea         | 22/06/2024 | 45.0752 | 33.6723 | (300, 0, 350, 500)       |
| EMIT | Black Sea         | 23/05/2024 | 45.0522 | 34.7289 | (0, 0, 50, 500)          |
| EMIT | Red Sea           | 29/06/2024 | 21.7277 | 36.5927 | (1100, 1300, 1150, 1800) |
| EMIT | Red Sea           | 21/06/2024 | 18.9457 | 37.8506 | (900, 590, 950, 1090)    |
| EMIT | Red Sea           | 21/06/2024 | 18.9457 | 37.8506 | (400, 1100, 450, 1600)   |
| EMIT | Red Sea           | 13/06/2024 | 16.0654 | 39.1138 | (1180, 500, 1230, 1000)  |

## Appendix D: Influence of zenith and azimuth angles on sun glint acquisition

400

405

In Fig. D1, we observe that there is a positive correlation between SZA and SGA for both missions. Regarding the acquired EnMAP data, most acquisitions have VZA values lower than  $20^{\circ}$ , which is translated into low sin (VZA) and high cos (VZA) values. According to Eq. 3, this would make the first term ( $\cos(\text{SZA}) \cdot \cos(\text{VZA})$ ) to contribute significantly more than the second one ( $\sin(\text{SZA}) \cdot \sin(\text{VZA}) \cdot \cos(\phi)$ ). Then, due to  $\cos(\text{VZA}) \sim 1$ , we can approximate  $\cos(\text{SGA}) \sim \cos(\text{SZA})$ , which explains the roughly linear relationship. This approximation is even stronger for the EMIT case (left column) because most VZA values are  $\sim 10^{\circ}$ , with minimal variation due to the instrument's lack of pointing capability. In addition, we find lower SGA values for acquisitions where VZA and SZA are similar and where  $\phi$  gets closer to  $180^{\circ}$  ( $\cos(\phi) = -1$ ), since these are the conditions to meet the angular configuration for sun glint. Due to the ability of EnMAP to point in the across-track direction, we find that EnMAP has more flexibility to achieve closer-to-sun glint acquisitions at the same SZA. For example, two points with a SZA of  $\sim 40^{\circ}$  have a SGA lower than  $20^{\circ}$  for EnMAP, while the minimum SGA value at this SZA is  $\sim 30^{\circ}$  for EMIT.

**Figure D1.** Scatter plots of SZA against SGA for the EMIT and EnMAP acquisitions from Table C1 and C2, showing the values related to the absolute difference between VZA and SZA and to the  $\cos(\phi)$ .

# Appendix E: Impact of the IA parameter in the scene radiance levels

In Fig. E1, we can see the Fresnel coefficient curve (see Eq. 5) related to the IA parameter, where the vertical lines are the minimum (Min) and maximum (Max)  $\rho$  values related to the GHGSat (orange), EnMAP (green), and EMIT (red) instruments. The GHGSat boundaries were extracted from the IA values from MacLean et al. (2024), while those from EnMAP and EMIT were obtained from the acquisitions from Table C1 and C2, respectively. For GHGSat, the absolute  $\rho$  difference between the Min and Max values (max( $\Delta \rho$ )) is 0.11, while for EnMAP and EMIT are around 0.01, which is an order of magnitude lower. Thus, the IA values will not have such an impact in EnMAP and EMIT compared to GHGSat.

**Figure E1.** Relationship between the IA parameter and the Fresnel coefficient with the minimum (Min) and maximum (Max) boundaries related to GHGSat (orange), EnMAP (green), and EMIT (red).

#### Appendix F: Retrieval precision to radiance conversion in the POD models

We relate  $\sigma_{\Delta XCH_4}$  to Rad by means of fits for EnMAP and EMIT data from Table C1 and C2. As shown in Fig. F1, we applied a power-law fit with  $R^2 = 0.962$  and an exponential decay fit with  $R^2 = 0.807$  for EnMAP and EMIT, respectively. Moreover,

in Fig. F2, we observe a similar plot to that in Fig. 10, but showing Rad instead of  $\sigma_{\Delta XCH_4}$ . To improve the visualization of the plots, we represented Rad in a log scale and overlay dashed white lines with constant Rad values.

Figure F1. Fitting curve relating  $\sigma_{\Delta XCH_4}$  to Rad for EnMAP (left) and EMIT (right) data from Table C1 and C2.

Data availability. Data will be made available on request.

Author contributions. Javier Roger: Conceptualization, methodology, formal analysis, investigation, writing (original draft), writing (review & editing). Luis Guanter: Conceptualization, methodology, writing (review), supervision. Javier Gorroño: Conceptualization, methodology, formal analysis, writing (review).

425 Competing interests. The authors declare that they have no conflict of interest.

Acknowledgements. We thank Daniel J. Varon for providing the WRF-LES simulations of methane plumes used in this study.

Figure F2. Rad dependency with Q and  $U_{10}$  extracted from EnMAP (top row) and EMIT (bottom row) data for (from left to right) POD  $\sim$  10 %,  $\sim$  50 %, and  $\sim$  90 %. EnMAP (points), PRISMA (crosses), and EMIT (squares) detections are overlaid with a Q value associated to the retrieval precision and  $U_{10}$  from the acquisition. Note that the Rad values are plotted in a log scale.

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
