# Peer review of "Assessing the Detection of Methane Plumes in Offshore Areas Using High-Resolution Imaging Spectrometers"

_EGUsphere, 2025_

## Author Comment (AC1)

**Reviewer 1**

L78 - The work presented here strongly relies on the quality of the retrieval method. What is the magnitude of the retrieval error? In particular, how does the retrieval error evolve with respect to the source rate? Please clarify this in the text. This is an important discussion as a high retrieval error could mean that some plumes might be detected with a more precise retrieval method for example. This directly impacts the estimation of Qmin.

*The retrieval error is measured with $\sigma_{\Delta XCH4}$, which is the standard deviation of the retrieval (Guanter, 2021). The number of pixels considered to deduce $\sigma_{\Delta XCH4}$ is so large that the presence of the plume and its related source rate has little effect. Moreover, the plume does not interfere with the retrieval since potential methane emission pixels are discarded to carry out the retrieval method (see L79-82 from original manuscript). We clarify this in the text.*

*We acknowledge that retrieval methodologies different from the one implemented in this work might lead to different results under the same detection algorithm. However, we use the matched filter since it is a state-of-the-art method that has been extensively used in the literature. We emphasize this in the text.*

*The magnitude of the retrieval error can be observed in Figure 7 (green dots). This is already explained in the text (Line 219-220 from the original manuscript)*

*Note that we also name $\sigma_{\Delta XCH4}$ as retrieval precision.*

L82 - It is common to measure the methane excess in ppm, however the total amount of gas also depends on the height of the column considered when estimating the concentration. Please add in the text the height of the column used for the retrieval. It will clarify the scale of Figure 6.

*We assume an integration over a 8-km column as in Thompson (2016). We include it in the text.*

L140 - Is only one band considered or a combination of bands?

*It is only one band. We change it to '… which we measure using the instrument radiance band located at …' for better clarity.*

L170 - How are the simulations included in the L1 acquisitions? In particular, how is chosen the plume location in the image ? Is the source placed where the sun glint is the most pronounced? What happens if only the tail of the plume is visible where there is sun glint?

- *The plumes are included in L1 data as in Guanter (2021). The methane enhancement retrieval map from the synthetic plumes are converted to transmissivity by means of a LookUp Table based on MODTRAN in spectral transmissivity mode. Next, this transmissivity is convolved to the instrument spectral response function. Then, the resulting transmissivity spectra is multiplied to the L1 data.*

  *Regarding this point, we think that 'as Guanter (2021)' should be enough for the reader to know how to implement simulations into L1 data.*

- *Simulated plumes were integrated into L1 data subsets listed in Table C1 and C2. These subsets are carefully selected by visual inspection to minimize the disturbance of surface structures that can lead to artifacts. Within each subset, we did not notice a significant change in the scattering glint angle parameter that would affect the emission detection. Then, we just implemented the synthetic plume in the upper part of the subset. As shown*

*later in Section 3, the whole subset selection provides a wide range of scattering glint angle values, which is valuable to understand the detection performance in reference to the sun glint configuration.*

**We do include this point in the text.**

- *Since our subsets do not show a significant change in the scattering angle parameter, we do not consider the case where the plume tail is not detected. Nevertheless, the detection algorithm used in this work is based on having a cluster of pixels higher than a certain threshold value. Generally, this cluster is located near the source and not in the plume tail.*

  *Regarding this point, we don't think we should include this information since we have already said that there is not a significant change of the scattering glint angle parameter within a subset. Therefore, we don't include this point.*

L189 - I appreciate that the authors acknowledged the limitations due to the dataset size. However, I believe that the biases linked to the small dataset size should be more extensively discussed. It is unlikely that 25 plumes can accurately represent all the possible scenarii for a given wind speed. However, this could be mitigated by a wise selection of the 25 plumes. Were the 25 plumes per wind speed sampled randomly or regularly during the LES procedure? What is the correlation between the samples? It might be interesting to compare the correlation between samples and their associated Qmin. To know if 25 samples are sufficient to estimate a POD I suggest to display the complete distribution of Qmin for different wind speeds.

*According to Ouerghi et al., (2025), depending on the time difference between adjacent plume instances, we have different levels of plume variability. At 30 s, there are some changes near the source, but not in the plume tail. However, at 60 s more noticeable changes occur. At 120 s, the plumes from adjacent instances are completely different. We used 11 groups of synthetic plumes (see Table 1 from original version), where 4 had a time difference of adjacent plumes between 30 s and 60 s, while 7 had a time difference higher than 60 s.*

[Figure]

*To indirectly test the impact of the time difference in the plume variability, we compare the Qmin variability of G1 and G2, which refer to the synthetic plume groups with U10 of 1.62 m/s and 1.74*

*m/s, respectively. As can be seen in this figure, these 2 groups have a time difference of 31 s and 46 s, respectively. Due to the relatively low difference in U10, we can assess the impact of the time difference in the plume variability. We use as a reference a scene with Rad = 0.1, which is translated into a noisier background where plume concentration plays an important role to pass our detection algorithm. Our results show that the standard deviation from G1 is 1.3 t/h, while the one from G2 is 2.3 t/h, which is translated into a 1 t/h difference. Therefore, we see that groups under very similar scenarios and conditions with a time difference between 30 s and 60 s can show a non-negligible difference in reference to plume variability. Due to the limited size of our simulation, we cannot set a time difference of 120 s for any group, which would be the ideal selection. However, we keep only those groups that exhibit a time difference higher than 60 s, which seems the best selection considering our dataset in order to increase plume variability as much as possible. Nevertheless, we acknowledge that there is room for improvement in this aspect.*

*Then, we remove the groups with a time difference lower than 60 s, and re-do calculations and figures that are affected by this. We also include some text regarding plume variability and room for improvement. We thank the reviewer for this comment since we consider that this is an important point.*

L184 - This means that a Qmin distribution is sampled for a fixed background. It could be interesting to estimate a Qmin distribution using several background with close Rad values. It will enhance the robustness of the Qmin estimate.

*We agree that adding more Rad values would enhance the robustness of the Qmin estimation. Adding more Rad values involves finding more offshore scenes captured by the EMIT and EnMAP instruments under favorable conditions (no clouds and free from artifacts). Nevertheless, as can be seen in Figure 7, an intensive sampling of Rad values have already been carried out. We think that this sampling should be representative enough as a first approach. In the text, we note that there is room for improvement regarding this point.*

L197 - In Eq.6 the parameters a and b are already used in Eq 2 with a different meaning. Please change those notations (or those in Eq 2).

*Corrected.*

L233 - In Fig 9, please clarify what the normalized NEdL curve brings to the figure.

*Corrected.*

L284 - The parameters a,b,c in Eq.7 are already used in Eq.2 and Eq.6 with a different meaning. Please change those notations.

*Corrected.*

---

## Author Comment (AC2)

**Reviewer 2**

1. Does XCH4 refer to the methane column concentration enhancement in Line 78?

*XCH4 refers to the methane column concentration, which can be more accurately named as 'dry-air column-averaged mole fraction' (Guanter, 2021). However, in Line 78 we write ΔXCH4, which refers to the methane column concentration enhancement. We are interested in the enhancement since this can be used to measure point source emissions. We change the term to 'methane column concentration enhancement' as suggested to enhance clarity.*

2. What are the errors of retrieval methane concentration from different instruments? How to consider the impact of background concentration on methane plume detection in Line 83?

*The retrieval methane concentration error is measured with $\sigma_{\Delta XCH4}$, which is the standard deviation of the retrieval (Guanter, 2021). The impact of background noise is captured with this magnitude and that is why $\sigma_{\Delta XCH4}$ is used in the detection algorithm. We emphasize this in the text.*

*Moreover, EMIT retrievals have generally lower $\sigma_{\Delta XCH4}$ than the ones from EnMAP. This can be shown in Figure 7 and is already explained in the text (Line 219-220 from the original manuscript).*

*Note that we also name $\sigma_{\Delta XCH4}$ as retrieval precision.*

3. How to consider the uncertainty of wind fields simulated by WRF-LES model in Line 106?

*The U10 related to a synthetic plume is obtained from the mean value of the U10 wind field from the simulated spatial domain covering a representative time before the plume instance time until the instance time itself. As a result, the extracted U10 value is representative of the plume instance methane distribution. However, no uncertainty related to this value is considered as in Varon et al., (2018).*

4. Why the U10 is not obtained from the WRF in Line 111? not consistent with the Ueff.

*Indeed, Ueff is extracted using the U10 values from WRF simulations. However, to quantify real emissions, we must extract a U10 value for the real emission area at the instrument acquisition time. We cannot extract this value from WRF simulations and therefore we extract this value from GEOS-FP.*

5. In Eq.6 and Eq.7, are the parameters same for the EnMAP and EMIT instruments?

*No. We change the notation to enhance clarity.*